# Learning the Visualness of Text Using Large Vision-Language Models

**Gaurav Verma**
Georgia Institute of Technology
gverma@gatech.edu

**Ryan A. Rossi**
Adobe Research
ryrossi@adobe.com

**Christopher Tensmeyer**
Adobe Research
tensmeye@adobe.com

**Jiuxiang Gu**
Adobe Research
jigu@adobe.com

**Ani Nenkova**
Adobe Research
nenkova@adobe.com

## Abstract

Visual text evokes an image in a person's mind, while non-visual text fails to do so. A method to automatically detect visualness in text will enable text-to-image retrieval and generation models to augment text with relevant images. This is particularly challenging with long-form text as text-to-image generation and retrieval models are often triggered for text that is designed to be explicitly visual in nature, whereas long-form text could contain many non-visual sentences. To this end, we curate a dataset of 3,620 English sentences and their visualness scores provided by multiple human annotators. We also propose a fine-tuning strategy that adapts large vision-language models like CLIP by modifying the model's contrastive learning objective to map text identified as non-visual to a common NULL image while matching visual text to their corresponding images in the document. We evaluate the proposed approach on its ability to *(i)* classify visual and non-visual text accurately, and *(ii)* attend over words that are identified as visual in psycholinguistic studies. Empirical evaluation indicates that our approach performs better than several heuristics and baseline models for the proposed task. Furthermore, to highlight the importance of modeling the visualness of text, we conduct qualitative analyses of text-to-image generation systems like DALL-E.

## 1 Introduction

People typically communicate knowledge and information textually, but most prefer to consume visually rich content. Text-to-image generation/retrieval models could augment text with appropriate images, aiding the creation of appealing and easy-to-understand documents. Models like DALL-E (Ramesh et al., 2022) and Stable Diffusion (Rombach et al., 2022) work phenomenally well for input text that is carefully constructed to elicit images. However, they cannot handle long-form text with a mix of sentences that may or may

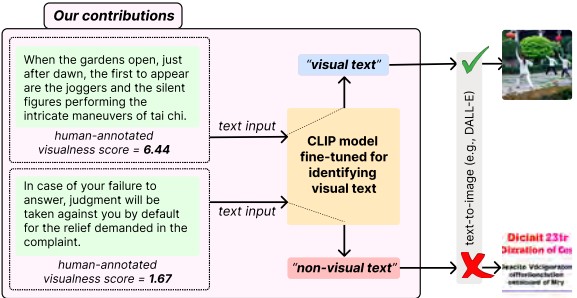

Figure 1: Overview of the sentence visualness identification task, along with a motivating downstream application (passive generation of relevant images).

not evoke a visual image. To this end, we introduce the task of identifying sentence *visualness*—a term we use interchangeably with *imageability*—as a necessary first step toward connecting long-form textual documents with relevant visual assets, without having to manually find visual sentences. In other words, to work effectively with long-form text without relying on manual input, text-to-image generation models like Stable Diffusion, DALL-E, and Imagen (Saharia et al., 2022) would benefit from inferring text visualness *before* they can generate images to embellish textual documents. In Figure 1, we demonstrate the need with some examples: text identified to have low visualness leads to irrelevant generations from DALL-E, while text identified to have high visualness leads to the generation of relevant images.

Prior approaches for quantifying the visualness of text operate on a word or phrase level (Deschacht and Moens, 2007; Jeong et al., 2012) and leverage lexicons that contain human-assigned world-level imageability scores (Louis and Nenkova, 2013). However, besides being limited in their coverage, our experiments also show that word or phrase-level visualness cannot be aggregated to quantify sentence-level visualness.

To this end, in this work, we curate a corpus of 3,260 sentences in English paired with their human

ratings for visualness, as well as a noisy but large corpus of 48,077 automatic alignments between text and visual assets in long-form documents. The textual part of the resulting alignment pairs can be used as examples of visual and non-visual sentences. We propose a strategy to fine-tune vision-language models like CLIP, allowing classification inferences over text-only inputs. Our objective also ensures that the learned embeddings remain usable for downstream text-to-image retrieval.

We compare the performance of our proposed approach against several heuristic and model-based baselines. Our extensive evaluation suggests that our fine-tuning strategy leads to the most accurate visual and non-visual text classifier. Finally, we conduct several analyses to glean insights into the model's learned attention mechanism, text-to-image retrieval abilities, and downstream text-to-image generation capabilities.[1]

In sum, our key contributions are:
• We propose the task of identifying the visualness of a sentence and curate a dataset by crowdsourcing annotations for English sentences.
• We develop a training objective that fine-tunes large vision-language models for the task of text visualness identification.
• Quantitative and qualitative experiments demonstrate the effectiveness of our fine-tuning approach in identifying visual text over several competitive baselines, while preserving downstream text-to-image retrieval performance.

## 2  Related Work

**Fine-tuning vision-language models for downstream tasks**: Large vision-language models like CLIP (Radford et al., 2021), UNITER (Chen et al., 2020), and ALIGN (Jia et al., 2021) have demonstrated remarkable performance on downstream tasks via transfer learning or fine-tuning. However, such downstream tasks assume *both* text and image as input to determine similarity or generate/retrieve the other modality for *every* instance of the corresponding modality; for instance, visual question answering (Antol et al., 2015), caption generation (Xu et al., 2015), and cross-modal retrieval (Wang et al., 2016). Fine-tuning large vision-language models on such downstream tasks involves adding components to the encoders' architecture and training additional parameters on

the task-specific dataset (Mittal et al., 2022; Sarto et al., 2022). Our work differs from existing work in that the input is only text, requiring us to adapt large vision-language models to not rely on both modalities during inference. We propose a fine-tuning strategy that does not involve additional architectural components (and parameters) on top of a pre-trained CLIP architecture and yet effectively adapts CLIP for learning text visualness. Our task can be considered a precursor to tasks like text-to-image retrieval and generation, where images are only retrieved or generated for visual text. Further, since reusability of representation is a desirable property (Yosinski et al., 2014; Long et al., 2015) we aim to preserve the reusability of text embeddings learned for the visualness categorization task for downstream tasks like text-to-image retrieval.

**Visualness of words**: The visualness of text has been studied in multiple prior works but at a word or phrase level. Coltheart (1981) curated the MRC Psycholinguistic Database comprising human ratings for imageability of 3769 words, which were later expanded using automated methods by Louis and Nenkova (2013). Beyond word-level visualness, some studies have focused on automated quantification of phrase-level visualness (Jeong et al., 2012; Deschacht and Moens, 2007). Our work focuses on learning sentence-level visualness instead of word or phrase-level visualness. While it is possible to aggregate word-level and phrase-level visualness scores to obtain sentence-level scores, it is unclear how accurate and generalizable these techniques are. We design multiple baselines that aggregate word-level scores to obtain sentence-level visualness and contrast the performance of such approaches with our proposed approach.

## 3  Text Imageability Dataset (TImeD)

Our proposed fine-tuning approach follows multi-stage training of a large vision-language model CLIP (Radford et al., 2021). In the first stage, we conduct large-scale fine-tuning, followed by fine-tuning on a relatively smaller annotated corpus in the second stage. We first discuss the curation of a large-scale corpus that comprises automatically-assigned and distant labels and then describe the curation of the human-labeled corpus of visual & non-visual sentences.

### 3.1  Fine-tuning with automatic labels

The formulation of the training objective (discussed later) requires positive examples comprising vi-

---

[1]Project webpage: `https://gaurav22verma.github.io/text-visualness/`

| Category | Example text from TIMED | $\mu$ / $\sigma$ |
|---|---|---|
| **Visual** | · now the snow has melted and the grass not only looks dreary, but it is soggy. | $\mu = 6.88$ |
| | · The operation left a six-inch zipper scar on his chest. | $\mu = 6.55$ |
| | · When the gardens open, just after dawn, the first to appear are the joggers and the silent figures performing the intricate maneuvers of tai chi. | $\mu = 6.44$ |
| | · He removed the box, placed it next to the garbage can, and put his garbage inside the can. | $\mu = 5.88$ |
| | · But, after running only the first 500 meters, he realized that the injury that seemed so insignificant would not only prevent him from winning the race, but also from finishing it. | $\mu = 5.00$ |
| **Non-visual** | · There's only one way to prove them wrong. | $\mu = 1.22$ |
| | · For more information or to schedule an outreach, please call (999) 123-4567 or email email@website.com. | $\mu = 1.55$ |
| | · In case of your failure to answer, judgment will be taken against you by default for the relief demanded in the complaint. | $\mu = 1.67$ |
| | · A 25% quorum of member votes in each district is needed to conduct district delegate elections in October. | $\mu = 1.77$ |
| | · Colliers International makes no guarantees, representations or warranties of any kind, expressed or implied, regarding the information including, but not limited to, warranties of content, accuracy and reliability. | $\mu = 2.00$ |
| **Ambiguous** | · J. Roman discusses his book Ohio State Football: The Forgotten Dawn which draws on extensive archival research to tell the untold story of the early days of football at Ohio as flagship public university. | $\sigma = 2.34$ |
| | · Remember to be sure to set your clocks back 1 hour before you go to bed on Saturday, November 3rd. | $\sigma = 2.23$ |
| | · That is the most important thing in my life today: Jesus. | $\sigma = 2.20$ |
| | · Children & parents will get to hear author George McClements read his book Ridin' Dinos with Buck Bronco. | $\sigma = 2.14$ |
| | · Financial Peace University is a nine-lesson class taught by financial expert Dave Ramsey through entertaining videos with an in-depth workbook, that will teach you how to take control of your money. | $\sigma = 2.16$ |

Table 1: Qualitative examples of visual and non-visual text from the human-annotated subset of the **Text Im**age**ability D**ataset (based on the average of annotator ratings), and text with high ambiguity (based on the standard deviation of annotator ratings).

sual text and paired images as well as negative examples that comprise non-visual text. To create a corpus like this, we: *(i)* leverage image-text co-occurrences in documents to develop a self-supervised approach, and *(ii)* use image-text similarity scores obtained using CLIP as priors to construct a large training corpus. We start with 450,000 publicly available PDFs referenced in the Common Crawl corpus and identify pages within those PDFs that include images.[2] We use a proprietary document object detection tool like Fitz[3] to extract paragraphs and images from the document pages.

We do sentence segmentation for the identified paragraphs using NLTK Tokenizer (Loper and Bird, 2002). To map the images in the page to sentences, we compute CLIP similarity scores between each image-sentence pair in a given page. Based on the distribution of image-sentence similarity scores across all the pages in our corpus, we set two thresholds, $T_{pos}$ and $T_{neg}$. A sentence in a page is considered a positive example (visual text) if its similarity with *any* of the images in the page is greater than $T_{pos}$. Similarly, chosen negative examples have similarity values less than $T_{neg}$ with *all* images within the same page. Sentences with an image similarity value greater than $T_{pos}$ are associated with the most similar image in the same page, while the negative examples are associated

with a common NULL image. The thresholds $T_{pos}$ and $T_{neg}$ are chosen conservatively to only include top or bottom $k$ % sentences from the entire corpus, respectively. This limits the noise in our training corpus for adapting the CLIP model for scoring text visualness. In our experiments, we set $T_{pos}$ to be $0.35$ to consider top $1\%$ sentences as visual and $T_{neg}$ to be $0.18$ to consider bottom $5\%$ sentences as non-visual. Our automatically-labeled corpus comprises 15,359 visual sentences, the corresponding images, and 32,718 non-visual sentences.

### 3.2 Human-annotated dataset

For the human-annotated visual and non-visual examples, we start with another 200,000 PDFs distinct from those used for the automated assignment of labels. To focus on natural images rather than infographics and academic figures, we filtered these documents to only include brochures, flyers, and magazines. For the resulting 35,432 documents, we adopted the same policy as that for curating the automatically-labeled dataset (selecting top 1% and bottom 5% sentences based on similarity values). We then recruited annotators to rate the visualness of the resulting 3,620 sentences after manually anonymizing any personal information.

We recruited annotators on Amazon Mechanical Turk (AMT). We randomly ordered the 3,620 examples and, for each example, we asked nine annotators to provide a response on a 7-point Likert scale for the following question: "*Do you agree that the sentence below evokes an image or picture in your mind?*" A response of 1 indicated strong disagreement, while 7 indicated strong agreement.

---

[2]We choose to work with PDF documents rather than webpages because *(i)* PDFs have natural demarcations in the form of pages (whereas webpages often contain long-running text with complex image-text interactions), and *(ii)* images within a page are likely to be related to selected text fragments within the same page.

[3]https://github.com/pymupdf/PyMuPDF

We also inserted some attention-check examples (5%; $n = 181$) to ensure the annotators read the text carefully before responding. These checks explicitly asked the annotators to mark a randomly chosen score on the Likert scale regardless of the actual content. We discarded the annotations from annotators who did not correctly respond to all the attention-check examples and re-collected more responses iteratively. Appendix A.3 provides more details about the filters used for recruiting the annotators and the annotation interface.

If a majority of annotations (i.e., at least 5 out of 9) were 1, 2, or 3, we considered the example to be non-visual ($n = 2108$). Similarly, visual examples had a majority of 5, 6, or 7 responses ($n = 1132$). We considered examples that did not have a clear majority or majority of responses of 4 (i.e., 'Neutral' on the Likert scale) as ambiguous and neutral, respectively. Table 1 shows illustrative examples of visual, non-visual, and ambiguous text from our human-annotated corpus.

For 27.1% of the examples only at most 1 of the 9 annotators disagreed with the labels decided based on the process described above. 10.5% of the sentences were assigned a neutral or ambiguous class. Inter-annotator agreement measured by Krippendorff's $\alpha$ was 0.446. This inter-annotator agreement value is in a similar range to what is observed for other language-related tasks that involve assessment of text by *experts* on dimensions like coherence, likability, relevance, and even grammar (Karpinska et al., 2021). For brevity, we refer to the curated dataset as TIMED, short for **T**ext **Im**ag**e**ability **D**ataset.

## 4  TIP-CLIP for Scoring Text Visualness

**Background**: The CLIP model (Radford et al., 2021) jointly trains image and text encoders to predict the correct pairing between images and textual descriptions. In a batch size of $N$ images and $N$ texts ($N^2$ possible image-text pairings), the objective function ensures that the cosine similarity between the embeddings of correct image-text pairs is maximized while the cosine similarity between the $(N^2 - N)$ incorrect image-text pairs is minimized. The encoders are trained over a large multimodal dataset of $\sim 400$ million image-text pairs.

**Updated training objective**: When predicting text visualness, the goal is to assign a higher score to text that is visual (evokes a concrete image for the person reading it) and a lower score for non-visual

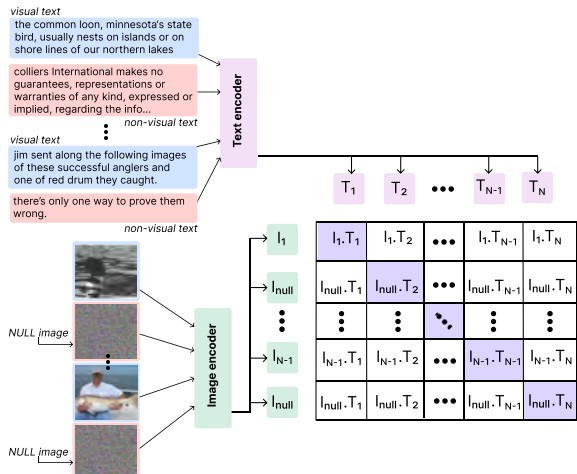

Figure 2: Our approach to predicting sentence visualness, with a fine-tuning strategy where visual text is matched with its corresponding image while non-visual text is matched with a fixed NULL image.

text (text that does not evoke an image). In line with the original training objective, we further train the CLIP model to match text that is identified as visual with the corresponding image. We adapt the CLIP training to match text that is identified as non-visual with a single NULL image (see Fig. 2). Matching visual text with the corresponding image while non-visual text to a NULL image not only encourages the model to distinguish between visual and non-visual text, but also allows it to anchor non-visual text in the common NULL image that can be used during inference without having access to a potentially paired image. Formally, the adapted training objective is given as,

$$\mathcal{L} = -\frac{1}{2N} \sum_{j=1}^{N} log \left( \frac{exp(\langle I_j^e, T_j^e \rangle / \tau)}{\sum_{k=1}^{N} exp(\langle I_j^e, T_k^e \rangle / \tau)} \right) -$$
$$\frac{1}{2N} \sum_{k=1}^{N} log \left( \frac{exp(\langle I_k^e, T_k^e \rangle / \tau)}{\sum_{j=1}^{N} exp(\langle I_j^e, T_k^e \rangle / \tau)} \right)$$

$$st. \; I_m^e = \begin{cases} I_{null}^e, & \text{if } m \in \bar{\mathcal{V}} \; (\text{i.e., non-visual}) \\ I_m^e, & \text{if } m \in \mathcal{V} \; (\text{i.e., visual}). \end{cases}$$
$$(1)$$

Here, $N$ denotes the number of examples in a batch, $I_m^e$ and $T_m^e$ denote the embeddings of the $m$-th pair of image and text that are normalized to have unit $\ell_2$-norm, respectively, such that $m \in \{1, \dots, N\}$. $\langle ... \rangle$ represents the inner product, and $\tau$ is the trainable temperature parameter. $\bar{\mathcal{V}}$ and $\mathcal{V}$ are the set of examples in the current batch

that belong to non-visual and visual categories, respectively. Finally, $I_{null}^e$ denotes the embedding of the NULL image. During inference, we compute the cosine similarity between the representation of a given text with the representation of the NULL image; non-visual texts will have a high similarity with the NULL image. Conversely, the visualness score $\mathcal{S}$ of any text with embedding $T^e$ can be obtained using

$$\mathcal{S} = 1 - \langle I_{\text{NULL}}^e, T^e \rangle. \tag{2}$$

For the NULL image, we create an RGB image of size $(224, 224, 3)$ in which each pixel value is chosen randomly (see Figure 2). However, experiments with different types of NULL images indicate that the choice of null image does not affect the model's performance; see Appendix A.1.

An alternative formulation for adapting the CLIP training objective could have been to match visual text with a single image while matching non-visual text with a single NULL image. However, this formulation of the training objective is similar to binary classification and does not enforce a contrastive objective for the positive examples. Matching visual text with its corresponding image instead of a common image for all visual text affords text embeddings that can be used for downstream tasks like text-to-image retrieval; we provide empirical evidence for worse text-to-image retrieval performance with the alternative formulation in Results.

## 5 Training details and Baselines

**Train, test, & validation splits**: Recall that our fine-tuning approach requires paired images for visual sentences only during training time and not during inference time; the model needs only text as input during inference. Of the 1132 visual sentences in the human-annotated set of TIMED, we assign 515 examples that had an automatically determined corresponding image to the training set, and the remaining were randomly assigned to the test set ($n = 517$) and validation set ($n = 100$). The 2108 non-visual sentences were randomly split into the training ($n = 980$), test ($n = 928$), and validation set (200). All three sets maintain positive:negative class ratio of $\sim 0.5$.

For the first stage of training, we fine-tune the CLIP model (ViT/B-32) on the proposed objective (see Eq. 1) using the 48,077 examples with automatic labels. This training is done on Tesla T4 GPUs, for 5 epochs, and a learning rate initialized at $5 \times 10^{-5}$ and optimized using Adam optimizer (Kingma and Ba, 2014). Following this, for the second stage, we further fine-tune the same model for 2 epochs using the same objective and hyper-parameters, but this time using the train set of human-annotated TIMED.[4] The hyper-parameters are selected by performing a grid search while observing performance on the validation set of TIMED. Based on the performance on the validation set of TIMED, we set the threshold of $\mathcal{S}$ (Eq. 2) to be 0.79 to categorize text as visual or non-visual. We refer to the model trained using our fine-tuning strategy as TIP-CLIP — **T**ext **I**mageability **P**redictor CLIP, and report performance on the test set of TIMED.

### 5.1 Baselines

We investigate the performance of TIP-CLIP against several heuristics and baseline models.

**Random**: The random baseline generates predictions via prior class probabilities in the training set.
**Average MRC-I score**: We consider the imageability scores of 3,769 words in the MRC lexicon and normalize them to be $\in [0, 1]$. For each example, we take the average of the imageability scores of the unique words; out-of-vocabulary words are assigned a score of 0. We lowercase the words in the MRC lexicon as well as the input text. Based on this average score, we categorize an example as visual or non-visual by setting the decision boundary as 0.17. The threshold is chosen to optimize performance on the validation set of TIMED.
**Concentration of Visual Genome objects (VG-Objects)**: The Visual Genome dataset comprises 75,729 objects, along with annotations for their attributes and object-object relations (Krishna et al., 2017). Based on the heuristic that a mention of a visual object in the text can trigger imageability, we quantify the concentration of Visual Genome objects by computing the fraction of unique object mentions in tokenized text with respect to the number of total unique words within the input text. We set the threshold to 0.5 based on the performance on the validation set.
**Expanding the MRC lexicon using word embeddings**: The coverage of the MRC lexicon is poor because it contains only 3,769 words. We expand this

---

[4]The CLIP model has a maximum context length of 77 tokens (about 50 words). Fewer than $1\%$ of the training examples are truncated to fit this context length.

list using semantic similarity between distributed representations of words (300-dim word2vec vectors trained on Google News corpus). For each word $w$ in the word2vec (Mikolov et al., 2013) vocabulary of pre-trained representations that does not occur in the MRC lexicon, we compute its cosine similarities with all the words in the MRC lexicon to identify the most semantically similar word that exists in MRC, given by $w_{\mathrm{MRC}}$ and its similarity with $w$ given as ($sim_{max}$). We assign the word $w$ an imageability score of $sim_{max} \times score_{w_{\mathrm{MRC}}}$, where $score_{w_{\mathrm{MRC}}}$ is the normalized imageability score of $w$'s most similar word $w_{\mathrm{MRC}}$. Based on the performance on the validation set, the decision boundary for average imageability score of input text is set as 0.17. This baseline propagation approach is highly effective in quantifying word-level imageability as the Pearson's correlation coefficient between the assigned visualness score and the average AMT rating of humans is 0.735 ($p < 0.001$); see Appendix A.2 for details.

**Fine-tuned BERT classifier**: We fine-tune a BERT model (`bert-base-uncased` on HuggingFace (Devlin et al., 2018; Wolf et al., 2020)) for the classification task of `visual` versus `non-visual` text detection. Similar to our proposed model, we adopt a two-stage fine-tuning approach with the BERT classifier (adding a classification layer to BERT for the first input token's (`[CLS]`) representation). We first fine-tune the model using the automatically labeled dataset followed by fine-tuning on the training set of the human-curated TIMED. For the first stage, we fine-tune the model for 7 epochs with a learning rate initialized at $5 \times 10^{-5}$ using a batch size of 32 while setting other hyper-parameters to default. We fine-tune the model for 3 epochs for the second stage with the same hyperparameters.

**Pre-trained CLIP model**: We use the pre-trained CLIP model (ViT/B-32) to obtain similarity scores between the embeddings of the `NULL` image (used for the fine-tuning of our model) and the input text. We then use $1 - \langle I_{\mathrm{NULL}}^{e}, T^{e} \rangle$ as an estimate of the visual score of text (see Eq. 2). Based on the performance on the TIMED validation set, we set the threshold for $\mathcal{S}$ to be 0.83.

## 6 Results and Analyses

**Evaluation on held-out test set of** TIMED: We first evaluate the baselines and our approach on the test set of the human-annotated TIMED, computing macro-averaged $F_1$, precision, recall scores,

| MODELS | $F_1 \uparrow$ | PRECISION $\uparrow$ | RECALL $\uparrow$ | ACC. $\uparrow$ |
|---|---|---|---|---|
| Random | 0.531 | 0.531 | 0.531 | 0.577 |
| MRC-I | 0.584 | 0.599 | 0.583 | 0.644 |
| VG-Objects | 0.606 | 0.610 | 0.605 | 0.646 |
| MRC-I + w2v | 0.638 | 0.637 | 0.639 | 0.667 |
| BERT | 0.753 | 0.766 | 0.789 | 0.756 |
| CLIP | 0.694 | 0.695 | 0.701 | 0.712 |
| **TIP-CLIP** (Ours) | **0.865** | **0.858** | **0.873** | **0.871** |

Table 2: Evaluation on human-annotated test set of TIMED. Reported $F_1$, Precision, and Recall values are macro-averages across the two classes (`visual` and `non-visual`).

and classification accuracy. Table 2 show the results for this evaluation. We observe that our proposed two-stage fine-tuning strategy leads to the best-performing model (TIP-CLIP). In comparison, the pre-trained CLIP model demonstrates notably weaker performance on the task of distinguishing visual text from non-visual text. Interestingly, fine-tuned BERT performs reasonably well on the task, considerably better than the CLIP model. Using the average imageability scores from MRC provides better-than-random performance but is severely subpar to models like CLIP, BERT, and TIP-CLIP. Using word2vec embeddings to expand the coverage of the MRC lexicon (i.e., MRC-I + w2v) leads to a boost in performance. However, collectively, the lacking performance of MRC-I and MRC-I + w2v demonstrates that word-level imageability does not translate to sentence-level imageability to a great extent. Notably, in terms of baselines that aggregate word-level attributes, VG-Objects provides the best estimate of sentence-level imageability by quantifying the concentrations of visual objects in the input sentence.

**Correlation of attention Weights with MRC imageability scores**: Attention mechanisms could be taken as proxies for explainability (Wiegreffe and Pinter, 2019; Chefer et al., 2021). Since the fine-tuned BERT, pre-trained CLIP, and our TIP-CLIP are attention-based models, we compute the correlation between average word-level attention scores (obtained from the last layer) on a given dataset with the imageability scores assigned by humans in the MRC lexicon. We compute these values for two datasets—the MSCOCO dataset (Vinyals et al., 2016) and the test set of TIMED. We only consider words that occur more than once in the specific corpus. Table 3 shows that TIP-CLIP attention scores correlate the most with MRC imageability scores,

| MODELS | MSCOCO | TIMED |
|---|---|---|
| BERT | 0.461*** (n = 344) | 0.326*** (n = 294) |
| CLIP | 0.448*** (n = 344) | 0.283*** (n = 294) |
| TIP-CLIP (Ours) | **0.497*** (n = 344) | **0.367*** (n = 294) |

Table 3: Correlation between MRC Imageability scores and model attention-scores for BERT, CLIP, and TIP-CLIP. $n$ denotes the number of overlapping words across vocabularies; *** denotes $p < 10^{-3}$.

| MODELS | $F_1$ ↑ | PRECISION ↑ | RECALL ↑ | ACC. ↑ |
|---|---|---|---|---|
| BERT (auto-labeled) | 0.714 | 0.704 | 0.716 | 0.710 |
| BERT (human-labeled) | 0.753 | 0.766 | 0.789 | 0.756 |
| BERT (auto + human-labeled) | 0.774 | 0.783 | 0.797 | 0.771 |
| CLIP | 0.694 | 0.695 | 0.701 | 0.712 |
| TIP-CLIP (auto-labeled) | 0.751 | 0.763 | 0.791 | 0.748 |
| TIP-CLIP (human-labeled) | 0.810 | 0.807 | 0.815 | 0.820 |
| TIP-CLIP (auto + human-labeled) | **0.865** | **0.858** | **0.873** | **0.871** |

Table 4: Ablation studies to understand the benefits of two-stage fine-tuning. The presented results are on the human-annotated test set of TIMED. Reported values are macro-averages of class-wise $F_1$, precision, and recall, and overall classification accuracy.

followed by the fine-tuned BERT's attention scores. The trends are consistent across both datasets. The relative ordering of models in terms of the correlation of their attention scores with MRC imageability scores follows the same order as their performance on the test set of TIMED. However, all correlation scores are in the low range, indicating a non-trivial relationship between sentence- and word-level imageability. The same trends hold for propagated visualness scores; see App. A.4. We also analyze the reason behind higher correlation scores on MSCOCO with respect to the TIMED corpus in Appendix A.4.

**Effect of multi-stage training**: We conduct ablations to isolate the effect of two-stage training. In Table 4, we show that BERT and TIP-CLIP can learn to distinguish `visual` and `non-visual` text even when fine-tuned only using the automatically labeled data. However, for both models, the gains from fine-tuning only on smaller, human-labeled data are notably higher. Furthermore, we find the proposed two-stage fine-tuning (i.e., training on automatically labeled data followed by human-labeled data) to be most effective, leading to a gain of over 2 and 5 absolute $F_1$ points over training only on human-labeled data for BERT and TIP-CLIP models, respectively. Additionally, for a given training strategy, our proposed fine-tuning of TIP-CLIP demonstrates better performance than the corresponding fine-tuned BERT model as well

as the standard pre-trained CLIP model.

**Effect on text-to-image retrieval**: We aim to analyze the re-usability of learned embeddings by the TIP-CLIP model for the text-to-image retrieval task. To this end, we consider the 515 `visual` examples from the test set of TIMED and, for each `visual` example, we rank the 515 corresponding images based on the cosine similarity between the image and text embeddings obtained from the TIP-CLIP model. We compute the Mean Reciprocal Rank (MRR) and contrast it with the MRR obtained using the pre-trained CLIP embeddings. As expected, CLIP achieves a near-perfect MRR of 0.989. The proposed fine-tuning objective does not severely impact the reusability of embeddings obtained from TIP-CLIP for retrieval, and results in an MRR of 0.937. This comparison evaluates the retrieval capabilities of TIP-CLIP against that of the CLIP model because the correspondence between visual text and images was established using similarities between CLIP embeddings.[5]

**The downside of an alternate training objective**: Recall that our fine-tuning strategy involves matching `visual` text with its corresponding image and matching `non-visual` text with the NULL image. With only the classification of `visual` and `non-visual` text in mind, an alternate fine-tuning strategy would have been to match all the `visual` examples with one common image while matching all the `non-visual` text with the common NULL image. The major downside of this approach is that while it leads to an effective classifier after two-stage fine-tuning, demonstrating a comparable $F_1$ score of 0.842 as the TIP-CLIP model, it performs poorly on the text-to-image retrieval task with an MRR of 0.014. Overall, while the alternate entirely classification-based training objective performs at par with the proposed TIP-CLIP model on the classification task, the resultant embeddings demonstrate poor reusability for downstream tasks like text-to-image retrieval.

**Properties of the new embedding space**: In Figure 3 we visualize the embedding space of the learned embeddings using t-SNE (Van der Maaten and Hinton, 2008). Alongside `visual` and `non-visual` sentences from the test set of TIMED,

---

[5]To automatically establish a correspondence between `visual` text and images, we enforce that the most similar image for a text should exist on the same page of the PDF. Therefore, it is possible that the CLIP similarity of text may be higher for a different image, resulting in an MRR slightly less than 1.0 (i.e., 0.989).

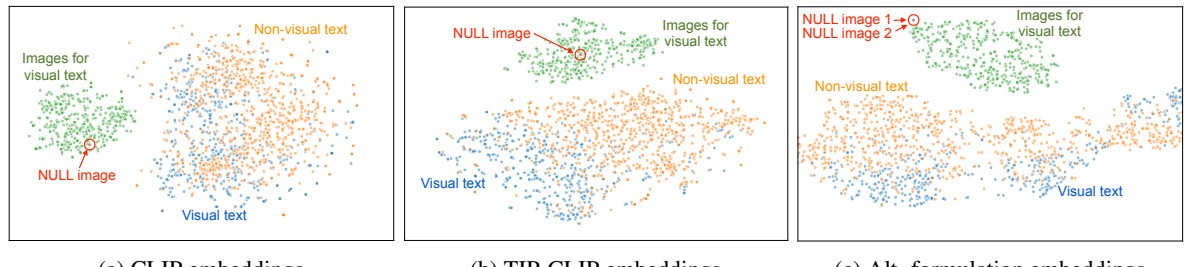

| (a) CLIP embeddings | (b) TIP-CLIP embeddings | (c) Alt. formulation embeddings |

Figure 3: t-SNE visualization of embeddings learned by (a) CLIP, (b) TIP-CLIP — using contrastive and adapted contrastive learning objective, respectively, & (c) model trained using alternative formulation solely focusing on classification. The plotted data points are from the TIMED test set. The observed "gap" in image & text spaces has been studdied by Liang et al. (2022).

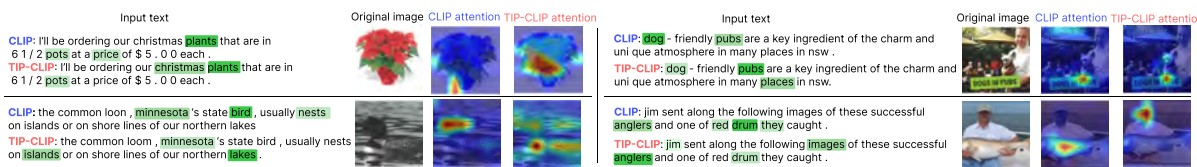

Figure 4: Comparing the attention maps over input text and images for CLIP and TIP-CLIP. For text, a darker shade of green demonstrates greater attention by the model. For images, red demonstrates the greatest attention in the heatmap. Image best viewed with zoom.

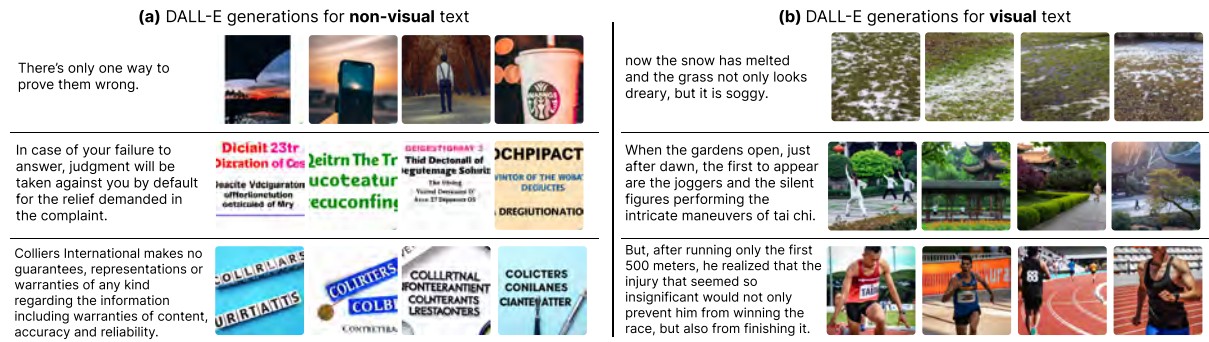

Figure 5: Examples of DALL-E generations for `non-visual` and `visual` text.

we also plot the embeddings of images corresponding to the `visual` sentences, and the embedding(s) of the `NULL` image(s). First off, we observe that the embeddings in Figure 3a and 3b from CLIP and TIP-CLIP are different in that the TIP-CLIP embeddings demonstrate better distinguishability between `visual` and `non-visual` text. In Figure 3c we observe that the alternative formulation pushes the `NULL` embeddings to the periphery of the image embeddings' cluster from a near-center location in Figures 3a and 3b. The text embeddings demonstrate notable distinguishability in Figure 3c too. We believe that the alternative classification-only formulation causes distortion in the latent space that causes drastic modification of text-only embeddings, making them useless for downstream text-to-image retrieval, as demonstrated empirically earlier. However, our proposed objective in TIP-CLIP pre-

serves reusability for downstream tasks by maintaining semantic relevance between learned image and text embeddings.

## 6.1 Qualitative Analysis

In this section, we conduct two qualitative analyses: *(i)* contrasting the attention mechanisms for CLIP and TIP-CLIP, and *(ii)* the role of distinguishing `visual` and `non-visual` text in downstream text-to-image generation using systems like DALL-E (Ramesh et al., 2021).

**Attention map visualization**: To contrast the mechanism by which CLIP and TIP-CLIP models match input text with their corresponding image, we visualize and contrast the attention maps for both models. We adopt the state-of-the-art approach to explain multimodal Transformers (Chefer

et al., 2021). In Fig. 4 we show 4 illustrative `visual` sentences from the test set of TIMED along with their corresponding images. Focusing on text, we observe that TIP-CLIP has a greater tendency to attend to visual aspects in the text; for instance, words like 'islands,' 'lakes,' 'anglers' are attended to a greater extent by TIP-CLIP than CLIP. In images, we observe small changes in attention maps across CLIP and TIP-CLIP; for instance, while the CLIP attention is focused on the Common Loon, TIP-CLIP also attends to the 'lake.' The qualitative analysis of visualization maps reinforces that the matching process for text and images undergoes small changes to accommodate greater attention to visual aspects in the text.

**Downstream text-to-image generation**: In Fig. 5 we show the generations obtained using DALL-E for text that is categorized as `non-visual` and `visual` in our dataset. We observe that for `non-visual` text, the images produced by DALL-E show poor relevance to the text. However, for `visual` text the generated images demonstrate great relevance to the input text.

Triggering text-to-image generation models like DALL-E for visual text is crucial to effectively use such systems in a passive setting. For instance, the authors should only be recommended to add visual assets in relevant places (i.e., for visual sentences) while working with long-form documents; triggering image generations for non-visual sentences could cause sub-optimal experiences. Thus, our contributions focus on distinguishing visual text from non-visual text as the necessary first step.

## 7 Conclusion and Future Work

We propose the task of predicting the visualness of text and curate a human-annotated dataset of sentence-level visualness scores. Additionally, we propose a two-stage fine-tuning objective for the task that involves training on a distantly supervised corpus followed by a smaller human-annotated corpus. Comparisons with several baselines demonstrate the effectiveness of our approach in distinguishing visual and non-visual text. We analyze the attention weights and downstream text-to-image retrieval capabilities of the model. Qualitative analysis of attention weights over textual input reinforces that our model attends to visual words to a greater extent. In closing, we show qualitative examples of how predicting text visualness can make text-to-image generation more effective.

In the future, we will study alternate objectives for learning text visualness while ensuring that the learned representations are transferable to related downstream tasks. We are also interested in using measures relating to the quality of the images generated from text-to-image generation systems to decipher signals about the visualness of input text, enabling the creation of auto-labeled examples. As the aggregation of word-level visualness scores leads to poor predictability of sentence-level visualness, future work could aim to understand what linguistic factors (like compositionality) precipitate sentence-level visualness.

## 8 Limitations and Broader Perspective

*Limitations*: As the first study on predicting sentence-level visualness, we focus on fine-tuning representative vision-and-language (CLIP) and language-only (BERT) encoders. Future studies can extend our experiments to explore the benefits of using other encoders to model text visualness. Our curated TIMED dataset only covers the English language. The notion of visualness can vary across languages and we encourage future research to contrast visualness in the context of the English language with that in other non-English languages. Additionally, since US-based crowd workers provided our ground-truth annotations for visualness, the dataset reflects a predominantly Western-centric view of text visualness. It is unclear how visualness in the text is perceived across different cultures. To this end, we acknowledge that our work and artifacts reflect West-centric views of visualness in the English language and encourage cross-lingual and cross-cultural extensions.

*Broader Social Impact, Annotations, and Datasets*: The authors do not foresee any negative social impacts of this work. However, our model can inherit the known biases in underlying models like CLIP and BERT (Agarwal et al., 2021; Garimella et al., 2021). The documents from which our datasets are curated are publicly available and are mentioned in The Common Crawl corpus (https://commoncrawl.org/); we abide by their terms of use. We manually anonymize instances of PII in the sentences that are annotated using Amazon Mechanical Turk and check for potentially offensive content. The recruited annotators are from the United States and are paid at an hourly rate of 12 USD.

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

## A Appendix

### A.1 Effect of the `NULL` Image

Since all the non-visual sentences in the training corpus are mapped to a common `NULL` image, we aim to see the effect of the chosen `NULL` image on the results. Recall that the `NULL` image used for our main experiments was obtained by creating an RGB image in which each pixel value is chosen randomly. We perform the same process with a different random seed to generate another `NULL` image. Additionally, we use a natural image as another alternative for the `NULL` image. These images are shown in Figure 6. We then evaluate the resulting models on the human-annotated test set of TIMED. Table 5 shows that the performance of the models is not dependent on the choice of the `NULL` image. We also find no dependence between the choice of the `NULL` image and the performance on downstream text-to-image retrieval.

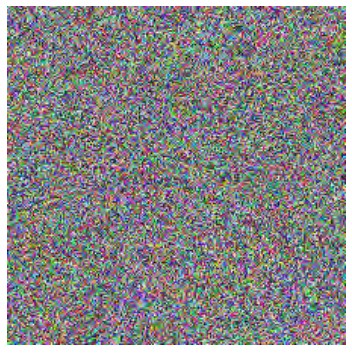 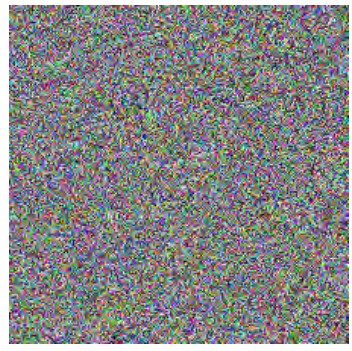 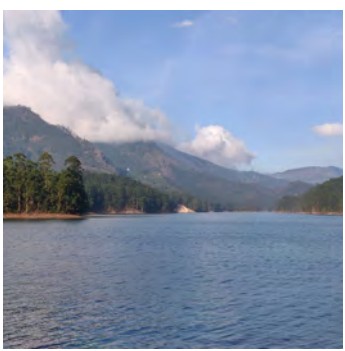

(a) Original NULL image        (b) NULL image with diff. seed        (c) Natural NULL image

Figure 6: Various NULL images used to study the effect of the chosen image on the text visualness identification task and the downstream text-to-image retrieval task.

| VARIANTS | $F_1$ ↑ | PRECISION ↑ | RECALL ↑ | ACC. ↑ | MRR ↑ |
|---|---|---|---|---|---|
| TIP-CLIP (Original – Fig. 6a) | 0.865 | 0.858 | 0.873 | 0.871 | 0.937 |
| TIP-CLIP (w/ diff. seed – Fig. 6b) | 0.867 | 0.854 | 0.875 | 0.872 | 0.934 |
| TIP-CLIP (natural image - Fig. 6c) | 0.861 | 0.855 | 0.876 | 0.872 | 0.939 |

Table 5: Effect of the choice of the NULL image on categorizing the human-annotated test set of TIMED and downstream text-to-image retrieval. Reported $F_1$, Precision, and Recall values are macro-averages across the two classes (visual and non-visual).

| Category | Example words (assigned score) |
|---|---|
| High imageability | martini, crabmeat, teeth, oysters, mosquitos, bracelets, motorboat, diamonds, squirrels, cigarettes, beaches, trumpets, dolphin, caramel, cattle, portobello, libraries, chimpanzee, snorkeling, sailboat, harmonica |
| Medium imageability | reassure, militancy, inhumanly, catalyses, industrial, peacefulness, handwoven, neurosurgery, overwashed, whooper, snails, preeminence, recluse, entrepreneur, character, insufficient, paladin, impersonal, deviously, recover |
| Low imageability | politologist, psycholinguistic, requirements, confirmatory, terseness, preformulation, offender, controversial, unhealable, monoculturalism, miserable, reprogrammability, this, participate, attractive, determinant, disestablishment |

Table 6: Qualitative examples of words that are assigned scores in the high ($\geq 0.7$), medium ($\in (0.3, 0.7)$), and low ($\leq 0.3$) range using the word2vec embedding-based propagation methodology.

## A.2 Assessment of word-level imageability score propagation

We randomly selected 500 words from the MRC lexicon and 500 words from the word2vec vocabulary that did not occur in the MRC lexicon. Each word was shown to 9 annotators using Amazon Mechanical Turk to seek responses to the following question: "*Do you agree that the word below evokes an image or picture in your mind?*" The annotators were instructed to respond on a 7-point Likert scale, where 1 denoted strong disagreement and 7 denoted strong agreement. Please see Appendix A.3 for details about the instructions, demographic filters, and compensation.

We average the ratings for all the annotated words and normalized them to be $\in [0, 1]$. We compute the Pearson's correlation coefficient between *(a)* the average ratings for MRC words and the normalized imageability scores, and *(b)* the average ratings for word2vec words and the imageability scores assigned via embedding-based propagation. The correlation between MRC imageability scores and average annotators' ratings is $0.870$ ($p < 0.001$) and the correlation between scores assigned via our propagation method and average annotators' ratings is $0.735$ ($p < 0.001$). This high positive correlation coefficient between assigned imageability scores and human-perceived ratings demonstrates the effectiveness of our adopted propagation method. We also note that the inter-annotator agree-

ments for the ratings for MRC words and word2vec words, as computed using Krippendorf's $\alpha$ (ordinal measure), were $0.626$ and $0.584$, respectively.

Overall, this assessment illustrates the validity of propagating word-level imageability scores using embedding-based semantic similarities. More broadly, the aim of adopting this approach is to expand the coverage of MRC lexicon. Qualitatively, we observe that words like 'gotcha' $(0.33)$ and 'presbyterian' $(0.61)$ are assigned meaningful imageability scores, demonstrating expansion along time and domains. As a point of difference between human ratings and assigned scores, we notice that the propagation approach assigned a high imageability score to words like 'qawwali' $(0.60)$ while the human annotators did not, possibly due to a lack of sociocultural context. In Table 6 we show illustrative words that are assigned high $(\geq 0.7)$, medium $(\in (0.3, 0.7))$, and low $(\leq 0.3)$ imageability scores using our propagation method.

### A.3 Details about MTurk Experiments

For all our annotation tasks, we recruited annotators using Amazon Mechanical Turk. We set the criteria to 'Master' annotators with at least a $99\%$ approval rate and were located in the United States. To further ensure the quality of annotations, we required the annotators to have at least $5000$ accepted annotations in the past. The rewards were set by assuming an hourly rate of $12$ USD for all the annotators. We show the annotation interfaces in Figure 7. In addition, the annotators were informed that the aggregate statistics of their annotations would be used and shared as part of academic research.

We also inserted some "attention-check" examples during the annotation tasks to ensure the annotators read the text carefully before responding. This was done by asking the annotators to mark a randomly chosen score on the Likert scale regardless of the actual content. We discard the annotations from annotators who did not correctly respond to all the attention-check examples and re-collect annotations for the affected samples.

### A.4 Further analyses on the correlation between attention scores and word-level visualness scores

We compute the Pearson's correlation coefficient between a model's average attention scores over words and the visualness score assigned using our propagation method. However, unlike Table 3, this time, we consider the propagated imageability

scores which lead to broader coverage in terms of vocabulary. As seen in Table 7, we observe the same trends as with MRC imageability scores, albeit with slightly lower values of correlation scores.

To analyze the alignment between learned attention scores for various models, we compute the correlation between average attention scores across different models. Pearson's correlation coefficients in Table 8 show that all the model attention scores have a moderate correlation with each other.

**Why are correlation scores higher for MSCOCO than for** TImED**?**: An interesting trend across Table 3 and 7 is that the correlation scores are consistently higher, across all the models under consideration, for the MSCOCO dataset than the test set of TImED. We note that, on average, MSCOCO has a caption length of $11.4$ whereas the TImED dataset has an average sentence length of $20.6$, with a greater concentration of objects from the Visual Genome objects—$6.7$ $(58.7\%)$ objects per example versus $8.4$ $(40.7\%)$ objects per example). For our TIP-CLIP model, these objects acquire an average of $63.2\%$ attention scores across all the MSCOCO examples, whereas they only acquire $37.1\%$ of attention scores, on average, across the examples in the TImED test set. Overall, these results demonstrate that the TIP-CLIP model attends over words in the MSCOCO corpus in an object-targeted manner but the attention is relatively diffused in the TImED corpus. Combined with the observation that MRC imageability scores are higher for concrete objects (Paivio et al., 1968), this explains why the correlation scores are consistently higher on MSCOCO than on TImED.

| MODELS | MSCOCO | TImED |
|---|---|---|
| BERT | 0.434*** | 0.301*** |
| CLIP | 0.429*** | 0.262*** |
| TIP-CLIP (Ours) | **0.465***** | **0.338***** |

Table 7: Pearson's correlation coefficient between propagated imageability scores (using word2vec) and model attention-scores. *** denotes $p < 0.001$

**Effect of length on the correlation between attention and MRC-I scores**: We categorize the sentences in the test set of TImED into short $(\leq 10; n = 304)$, medium $(\in (10, 20); n = 505)$, and long $(\geq 20; n = 606)$ sentences based on word counts. However, we did not find a notable variation in the correlation scores between the attention weights of the TIP-CLIP model and MRC Imageability scores. Pearson's correlation coeffi-

*(a) Interface to collect sentence-level visualness scores*

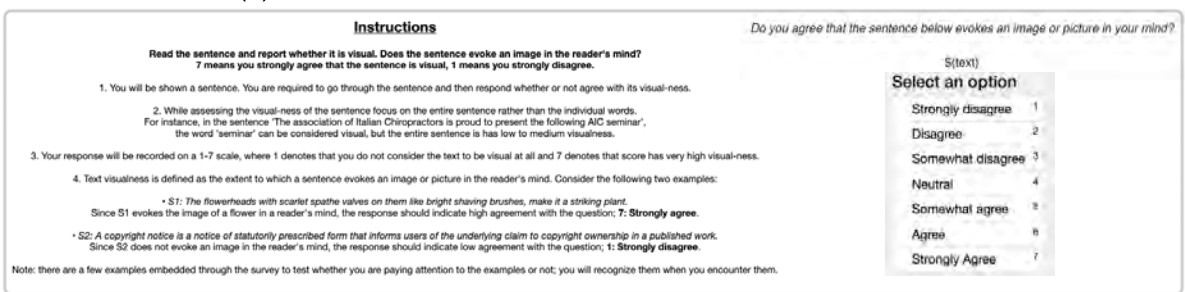

*(b) Interface to evaluate word-level visualness scores assigned by the propagation method*

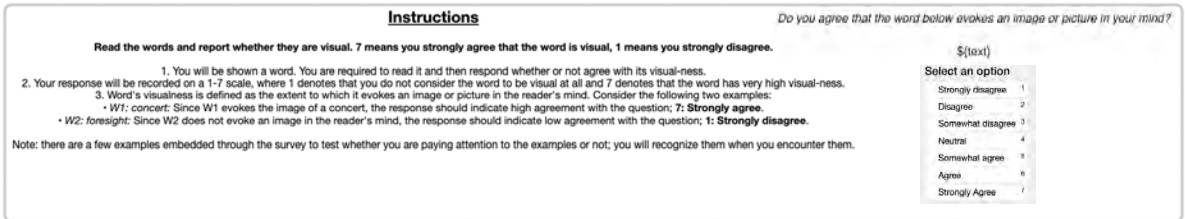

Figure 7: Interface for our annotation tasks on Amazon Mechanical Turk. For each of the annotations task, we also show the instructions provided to the annotators.

| MODELS | BERT | CLIP | TIP-CLIP |
|---|---|---|---|
| BERT | — | – | – |
| CLIP | 0.552*** | – | – |
| TIP-CLIP (Ours) | 0.631*** | 0.571*** | – |

Table 8: Pearson's correlation coefficient between word-level attention scores of various models for the TIMED test set. *** denotes $p < 0.001$

| MODELS | $F_1$ ↑ | PRECISION ↑ | RECALL ↑ | ACC. ↑ |
|---|---|---|---|---|
| Random | 0.503 | 0.503 | 0.503 | 0.505 |
| MRC-I | 0.470 | 0.472 | 0.472 | 0.470 |
| VG-Objects | 0.536 | 0.541 | 0.539 | 0.548 |
| MRC-I + w2v | 0.501 | 0.502 | 0.504 | 0.502 |
| MRC-I + GloVe (Twitter) | 0.516 | 0.518 | 0.520 | 0.519 |
| BERT | 0.612 | 0.634 | 0.624 | 0.618 |
| CLIP | 0.644 | 0.645 | 0.645 | 0.644 |
| **TIP-CLIP** (Ours) | **0.696** | **0.693** | **0.691** | **0.694** |

Table 9: Out of domain evaluation on the Twitter dataset. Reported $F_1$, Precision, and Recall values are macro-averages across the two classes (`visual` and `non-visual`).

cient was 0.33, 0.35, and 0.37 for short, medium, and long sentences, respectively. We observed the same trend for the fine-tuned BERT model and the pre-trained CLIP model.

### A.5 Out-of-Domain Generalization

Robustness of vision-language models has been the subject of investigation in several prior works (Verma et al., 2022; Ramshetty et al., 2023; Li et al., 2021). A critical assessment of the robustness and generalizability of the models trained

using our proposed approach is to conduct evaluations on out-of-domain (OOD) datasets. To this end, we curate a social media dataset by scraping Twitter. We start with the Wikipedia-based Image Text Dataset (WIT) (Srinivasan et al., 2021) and query Twitter using the Wikipedia page title to retrieve posts in English that are *with* and *without* images. We require that the retrieved post contains the page title string to ensure topical similarity between posts with and without images. To remove examples with irrelevant images, we discard posts with a CLIP-similarity lower than 0.70 between the Twitter post's image and the corresponding image on Wikipedia. Consequently, we obtain a dataset of Twitter posts containing mentions of 1185 Wikipedia topics, 7844 Twitter posts with images, and 7248 Twitter posts without images. The posts with and without images are tied by common Wikipedia topics.

We hypothesize that the text in Twitter posts that mention a certain topic and contain an image is more visual than text in Twitter posts that mention the same topic and do not contain any images. To test this hypothesis, we randomly sample 40 Wikipedia topics and present the associated text with ($n = 264$) and without images ($n = 241$) to human annotators. In an AMT survey that follows the design for curating TIMED, we find that the average annotator rating for the text from Twitter posts *without* images is 2.306 ($\pm 1.369$) while that for text from Twitter posts *with* images is

4.304 ($\pm 1.273$). We observe the inter-annotator agreement of $0.413$, which is similar to that observed while curating TIMED. For 34 out of the 40 Wikipedia topics, the annotators provided a higher imageability rating to text originally associated with an image on Twitter than text not associated with an image. Overall, the AMT survey validates our hypothesis by demonstrating that text in Twitter posts with images is perceived as more visual than text in Twitter posts without images, modulo the topic is common across the posts.

We now ask the question: how well the models considered in our work categorize Twitter text with images as `visual` and Twitter text without images as `non-visual`? We first adapt the thresholds used to classify text using various methods by running an evaluation on a randomly sampled validation set of 100 Twitter examples, 50 from each category. The thresholds are set as follows: MRC-I: 0.19; VG-Objects: 0.52; MRC-I + w2v: 0.17; MRC-I + GloVe: 0.32[6]; CLIP: 0.87; TIP-CLIP: 0.74. Using these threshold values, we categorize the rest of the Twitter dataset ($n = 14,992$) into visual and non-visual categories. The random baseline uses uniform sampling.

Table 9 shows the results for this out-of-domain evaluation. First, we note that all models undergo a severe drop in performance on the OOD dataset, indicating that the notion of sentence-level imageability is strongly tied to the domain. Our proposed TIP-CLIP model demonstrates better OOD generalization capabilities than all the considered baselines. It is noteworthy that the fine-tuned BERT model performs poorly on the OOD dataset than the standard pre-trained CLIP model. The aggregation of word-level imageability scores provides a worse-than-random estimate of sentence-level imageability on the OOD dataset.

### A.6 Predictions on Ambiguous Sentences

Recall that while curating TIMED, we combined examples without a clear majority from the annotators ($n = 378$) and those with majority votes for the 'Neutral' category ($n = 2$) into a single category called `ambiguous`. We revisit these examples to analyze how the most competitive baselines

---

[6]Since we are operating with the Twitter domain, we design a version of the propagation method where MRC Imageability scores are propagated in the GloVe-embedding space, where the GloVe embeddings are learned on Twitter corpus (Pennington et al., 2014). We use 200-dimensional GloVe vectors trained on 2 billion Twitter posts with a vocabulary size of 1.2 million.

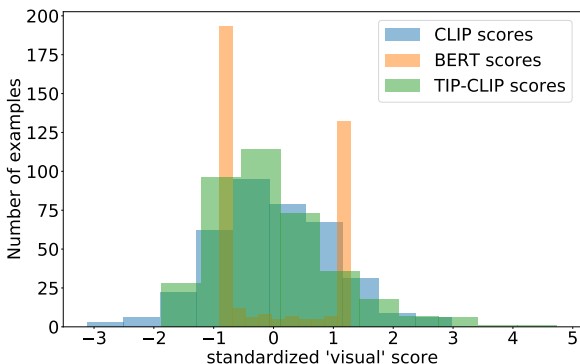

Figure 8: Distribution of standardized visualness scores for ambiguous examples (i.e., $(v - \mu)/\sigma$, where $v$ is the original visualness score, $\mu$ and $\sigma$ are the mean and standard deviation of the distributions, respectively). We contrast the predicted visualness scores by fine-tuned BERT, pre-trained CLIP, and our TIP-CLIP models.

and our proposed TIP-CLIP model score them on imageability. We compute the imageability score using Equation 2 for CLIP and TIP-CLIP, while treating fine-tuned BERT's prediction probability score as its imageability score for a given example. To appropriately compare the distribution of imageability scores across these three models, we standardize the values by computing $z$-scores (i.e., $x_i$ is transformed into $z_i = (x_i - \mu)/\sigma$; where $x_i$ is the original value, $\mu$ and $\sigma$ are mean and standard deviation of the distribution that $x_i$ belongs to). In Figure 8, we show that while CLIP and TIP-CLIP imageability scores are distributed normally around their respective means, BERT imageability scores are bimodal with peaks close to one standard deviation away from their mean. This demonstrates that if the models were to be used for *scoring* text imageability, as opposed to *categorizing* text into `visual` and `non-visual` categories, CLIP and TIP-CLIP models will provide more reasonable middle-level scores for `ambiguous` text, whereas scores from BERT would either be higher or lower. We attribute this to how the underlying models are trained and how the consequent imageability scores are computed. While the BERT model is trained solely for the classification task that emphasizes discriminative encoding and the predicted probability score is used as the imageability score, the distribution is bimodal. However, CLIP and TIP-CLIP are trained using image-text matching (the former, entirely; the latter, to some extent), and imageability scores are computed as the distance between the `NULL` image and input text.