# OpenReview forum: "Learning the Visualness of Text Using Large Vision-Language Models"
_EMNLP/2023/Conference — EMNLP 2023 Main_

### Official Review · Reviewer_3qDT · 2023-08-01

**Soundness:** 4

**Excitement:**

3: Ambivalent: It has merits (e.g., it reports state-of-the-art results, the idea is nice), but there are key weaknesses (e.g., it describes incremental work), and it can significantly benefit from another round of revision. However, I won't object to accepting it if my co-reviewers champion it.

**Paper Topic And Main Contributions:**

The objective of this paper is to determine the visualness of sentences, for which it creates a dataset by gathering annotations through crowdsourcing for English sentences. To accomplish this, it devises a training objective that fine-tunes large vision-language models specifically for identifying the visual aspect of text. Through both quantitative and qualitative experiments, it shows the remarkable effectiveness of its fine-tuning method compared to various competitive baselines in recognizing visual text, all the while maintaining the performance of downstream text-to-image retrieval.

**Reasons To Accept:**

1. This paper introduces a novel task aimed at identifying the visualness of a sentence.
2. To achieve this objective, this paper compiles a corpus consisting of 3,260 English sentences, each paired with human ratings for visualness. Additionally, it gather a noisy yet extensive corpus of 48,077 automatic alignments between text and visual assets found in long-form documents.

**Reasons To Reject:**

1. The overall size of the corpus is relatively small, especially the high-quality portion that is manually annotated.
2. The innovation is somewhat lacking, and the proposed fine-tuning strategy is relatively simple.
3. No mention about publishing the code and dataset.

**Reproducibility:**

4: Could mostly reproduce the results, but there may be some variation because of sample variance or minor variations in their interpretation of the protocol or method.

**Reviewer Confidence:**

4: Quite sure. I tried to check the important points carefully. It's unlikely, though conceivable, that I missed something that should affect my ratings.

---

> ### Author Rebuttal · Authors · 2023-08-28
>
> We thank the reviewer for their comments. In this rebuttal, we present counter-arguments for the reviewer’s stated reasons for rejection: the small size of the manually annotated corpus and the simplicity of the approach, and provide details about code and data sharing.
>
> **Relatively small-sized manually annotated corpus**: The manually annotated corpus (n = 3,620) is not small compared to other manually annotated corpora that are now part of standard benchmark datasets. For instance, the GLUE benchmark (Wang et al., ICLR 2019) includes STS-B and RTE datasets that have the same order of examples. The same applies to the AdvGLUE benchmark (Wang et al., NeurIPS 2021), with 5 tasks with fewer manually verified examples. Additionally, large vision-language and language models demonstrate lesser dependence on fine-tuning corpus size. We believe the manually annotated corpus is appropriately sized to enable effective training and evaluation for the proposed task. We empirically demonstrate this in Table 4, where the human-labeled data alone can be used to train a state-of-the-art TIP-CLIP model without using the auto-labeled data.
>
> **The simplicity of the approach**: The proposed fine-tuning approach is intentionally simple and yet effectively performs the task when compared to the baselines. We believe that effective and simpler approaches are preferred as they are less brittle and easy to deploy in real-world settings.
>
> **Code and data-sharing**: We mention in lines 698-700 that we will release the human-annotated data and code associated with our work to aid reproducibility and encourage future work on the topic. Upon acceptance, we will include the links to the data and code repositories.

---

### Official Review · Reviewer_dbaf · 2023-08-04

**Soundness:** 3

**Excitement:**

4: Strong: This paper deepens the understanding of some phenomenon or lowers the barriers to an existing research direction.

**Paper Topic And Main Contributions:**

The paper "Learning the Visualness of Text Using Large Vision-Language Models" addresses the problem of identifying the visualness of text, which is the ability of a text to evoke an image or picture in a person's mind. This is particularly important for tasks such as text-to-image retrieval and generation, where models like DALL-E and Stable Diffusion can benefit from inferring text visualness before generating images to embellish textual documents.

The main contributions of the paper are as follows:
1. The authors propose the task of identifying the visualness of a sentence and curate a dataset by crowdsourcing annotations for English sentences
2 They develop a training objective that fine-tunes large vision-language models, such as CLIP, for the task of text visualness identification
3. Quantitative and qualitative experiments demonstrate the effectiveness of their fine-tuning approach in identifying visual text over several competitive baselines, while preserving downstream text-to-image retrieval performance.


**Questions For The Authors:**

1. In the paper, DALL-E was used as a tool for quantitative analysis experiments, providing a small amount of generated content for both visual and non-visual text. However, it seems that there is no visual quantification of the generated images for the TIMED test set and out-of-domain data. It is important to ensure that the fine-tuning strategy for the two stages of CLIP not only optimizes the classification performance of visual and non-visual sentences, but also ensures the generalization performance of CLIP and the generation performance of text-to-image under an open vocabulary setting. Additional explanations may be needed.

2. When doing filtering, CLIP sets Tpos to 0.35 and Tneg to 0.18. Based on previous experience, a filtering threshold of 15% is already high enough for CLIP. Why did you adopt the positive sample filtering strategy of 0.35 and the negative sample threshold strategy of 0.18? Is it because taking only the bottom and top parts makes this threshold more effective?

3. Although the appendix mentions the possibility of conducting a differential analysis of text visual entailment at the word and sentence levels in the future, would there be any consideration for conducting a small experiment to support the importance of the superior sentence-level mentioned in the paper？

**Reasons To Accept:**

Strengths:
1. Novel task: The paper introduces a new task of identifying the visualness of text, which is important for various applications like text-to-image retrieval and generation.
2. Dataset creation: The authors curate a dataset with human annotations for visualness in English sentences, which can be useful for future research in this area.
3. Fine-tuning approach: The paper proposes a fine-tuning strategy for large vision-language models like CLIP, enabling them to identify visual text while preserving downstream performance.
4. Experimental validation: The authors provide quantitative and qualitative experiments to demonstrate the effectiveness of their approach compared to competitive baselines.

**Reasons To Reject:**

The paper focuses on the contrastive learning paradigm between visually rich and visually impoverished text, which can better uncover which visual information in text is truly aligned with images. However, one weakness of this paper is that only a small amount of quantitative analysis was conducted using generative models such as DALL-E，which is not sufficient to prove that the text-visual semantic discrimination method can effectively help downstream task scenarios, such as image-text retrieval and text-based image generation quality.


**Reproducibility:**

3: Could reproduce the results with some difficulty. The settings of parameters are underspecified or subjectively determined; the training/evaluation data are not widely available.

**Reviewer Confidence:**

3: Pretty sure, but there's a chance I missed something. Although I have a good feel for this area in general, I did not carefully check the paper's details, e.g., the math, experimental design, or novelty.

---

> ### Author Rebuttal · Authors · 2023-08-28
>
> We thank the reviewer for their constructive comments. We also thank the reviewer for noting the novelty of the task and the usefulness of the curated dataset for future research. In the rebuttal, we answer the questions and provide additional clarification for the motivation of the work.
>
> **Motivation of the work**: The reviewer mentions that the analysis conducted using generative models such as DALL-E is insufficient to prove that identifying text visualness can effectively help downstream image retrieval and generation quality. We wish to clarify that the motivation of the work is to identify text visualness as the first step of the process that could connect long-form documents with visual assets. Once the visual text has been identified, existing text-to-image retrieval and generation methods can be triggered just for visual text. To this end, our contributions are orthogonal to the motivation of improving image retrieval and generation quality. Instead, we argue that retrieving and generating images only for visual text will enable a more efficient and targeted usage of these models, as non-visual text seldom needs visual assets to be retrieved or generated. However, our analysis demonstrates that once the CLIP model has been fine-tuned to identify visual text from non-visual text, it preserves the retrieval capabilities of the pre-trained CLIP model. Additionally, in Figures 4 and 6, we qualitatively show how DALL-E generates absurd images for non-visual text, demonstrating the reason why distinguishing visual and non-visual text is beneficial.
>
>
> **(A1) Question about the images in the TIMED test set**: It seems that the reviewer considers that we used DALL-E to generate images for visual and non-visual text in our data. We wish to clarify that this is not the case. To construct the large auto-labeled training corpus, the text and the images were taken from publicly available PDFs using CLIP-based similarity scores (heuristic: visual text is likely to have higher similarity with the image within the same PDF page, whereas non-visual text is likely to be dissimilar to the image). Recruited crowd workers manually annotated a subset of this large-scale auto-labeled data, which makes up the TIMED train and test set (please see Sections 3.1 and 3.2). None of the content was generated using DALL-E. To clarify further, in the analysis, Figures 4 and 6 aim to demonstrate how DALL-E generates absurd images for non-visual text and relevant images for visual text, thereby motivating why the distinction between visual and non-visual text is important.
>
> **(A2) Clarification question about the thresholds for filtering the auto-labeled data**: We thank the reviewer for this insightful question. As the reviewer notes, the reason why the thresholds T_pos and T_neg were set to 0.35 and 0.18, respectively, is to limit the noise in the auto-labeled data. We only consider top and bottom k % sentences (based on similarity scores) to obtain about 15,000 positive and 32,000 negative examples (lines 197-207).
>
> **(A3) Clarification about the importance of sentence-level visualness**: We believe both word-level and sentence-level visualness are important, depending on the application. For instance, if the subject of analysis is web search queries, modeling visualness at a word or phrase level would be more beneficial. However, for long-form documents, which is the focus of our work, modeling sentence-level visualness will be more direct. In Table 2, we show that aggregating word-level visualness is not an effective strategy to predict sentence-level visualness – please see MRC-I, VG-Objects, and MRC-I + w2v baselines. In the discussion of future work, we indicate that this could be due to compositionality (lines 658-662) and could be the focus of a future investigation. While existing works have studied word- or phrase-level visualness (please see ‘Visualness of words’ under Related Work), our work focuses on modeling sentence-level visualness.
>
> Regarding **reproducibility**, we mention in lines 698-700 that we will release the code and human-annotated data upon acceptance. We will include the links to the data and code repositories in upcoming versions of the paper.

---

### Official Review · Reviewer_B9nx · 2023-08-11

**Typos Grammar Style And Presentation Improvements:** Figure 4-A
**Soundness:** 3

**Excitement:**

4: Strong: This paper deepens the understanding of some phenomenon or lowers the barriers to an existing research direction.

**Missing References:**

I didn't identify any missing references.

**Paper Topic And Main Contributions:**

In this paper, the authors propose the task of predicting the visualness of a given text. They also propose a fine tuning strategy that utilize large visual language models to solve the problem. Moreover, a dataset is curated and published for the visualness problem. The method is explained in detail, experiments and results are provided.

**Questions For The Authors:**

Question A: Why did you chose BERT-base over BERT-large as your baseline?
Question B: Latest cited work on visualness in the paper dates back to 2013. Wondering whether there aren't any other work since then?

**Reasons To Accept:**

The paper is very well written. The task of identifying the visualness of a text phrase is defined and a relevant solution is proposed along with a dataset.

**Reasons To Reject:**

A stronger baseline (such as BERT-large instead of BERT-base) could be selected.

**Reproducibility:**

4: Could mostly reproduce the results, but there may be some variation because of sample variance or minor variations in their interpretation of the protocol or method.

**Reviewer Confidence:**

3: Pretty sure, but there's a chance I missed something. Although I have a good feel for this area in general, I did not carefully check the paper's details, e.g., the math, experimental design, or novelty.

---

> ### Author Rebuttal · Authors · 2023-08-28
>
> We thank the reviewer for their thoughtful comments and feedback. In this rebuttal, we present some results that demonstrate that BERT-base is a good representative baseline model for text-only classifiers for the task of visual text identification. Below, we also show the performance of the RoBERTa (base and large) encoders (Liu et al., 2019) for an exhaustive comparison with our proposed TIP-CLIP model.
>
> | Model                              | F1 Score | Precision | Recall | Accuracy |
> | ---------------------------------- | -------- | --------- | ------ | -------- |
> | BERT-base (auto + human-labeled)    | 0.774    | 0.783     | 0.797  | 0.771    |
> | BERT-large (auto + human-labeled)   | 0.786    | 0.791     | 0.799  | 0.782    |
> | RoBERTa-base (auto + human-labeled) | 0.794    | 0.798     | 0.801  | 0.786    |
> | RoBERTa-large (auto + human-labeled)| 0.801    | 0.799     | 0.811  | 0.793    |
> | TIP-CLIP (auto + human-labeled)     | 0.865    | 0.858     | 0.873  | 0.871    |
>
>
> Since the performance on our task is similar with BERT-base and BERT-large models and since the overall trends remain the exact same with both the model variants as well as with the RoBERTa encoders (please refer to Tables 2 and 4 for complete results), we perform the subsequent experiments in the paper with a more efficient (in terms of training time and cost) model – i.e., BERT-base. As per the reviewer’s suggestion, we will include this information in the appendix.

---

### Official Review · Reviewer_4r5P · 2023-08-11

**Soundness:** 3

**Excitement:**

3: Ambivalent: It has merits (e.g., it reports state-of-the-art results, the idea is nice), but there are key weaknesses (e.g., it describes incremental work), and it can significantly benefit from another round of revision. However, I won't object to accepting it if my co-reviewers champion it.

**Paper Topic And Main Contributions:**

This paper curate a dataset containing English sentences and their visualness score. It also proposes a fine-tuning strategy to identify whether a text is visualness and its corresponding image.

**Reasons To Accept:**

(a) The paper propose a new task of identifying the visualness of a sentence and curate a new dataset accordingly.

(b) The experiment is comprehensive and the result analysis is detail.

**Reasons To Reject:**

(a) The paper lacks clarity in demonstrating the significance of its work and fails to outline the potential benefits that recognizing visual information in text could bring to other applications. As for the image generation model mentioned in the introduction, it can generate reasonable images for visual text. However, even if the visualness of the text is recognized, it cannot change the quality of generated images. Therefore, there's a need for further clarification regarding the importance of this task.

(b) The utilization of the pre-trained CLIP model as a baseline in Line 443 raises concerns. Representing the visual score of text through the 1-<I, T> measure seems unreasonable. While a high score of 1-<I, T> indicates low similarity with a NULL image, it doesn't necessarily correlate to high similarity between the text and an actual visual image.

**Reproducibility:**

4: Could mostly reproduce the results, but there may be some variation because of sample variance or minor variations in their interpretation of the protocol or method.

**Reviewer Confidence:**

3: Pretty sure, but there's a chance I missed something. Although I have a good feel for this area in general, I did not carefully check the paper's details, e.g., the math, experimental design, or novelty.

---

> ### Author Rebuttal · Authors · 2023-08-28
>
> We thank the reviewer for their comments. We clarify the two aspects of the work the reviewer mentioned: the significance of the work and the use of the CLIP model as one of the baselines.
>
> **Significance of the work**: The reviewer mentions that for the work to be considered significant, it needs to demonstrate better quality of images generated by text-to-image models. We believe this to be orthogonal to our key contributions in the work. We state that identifying text visualness is a necessary _first step_ toward connecting long-form textual documents with relevant visual assets, which could be done using retrieval or generation as the subsequent steps (lines 43-50). This motivation is independent of the image-generation capabilities of the models like DALL-E. As an example, let us consider that there are 1000 sentences in a long-form document. If the document can be enriched using 5 or 10 images, it would help to know which portions of the document are visual and which are not. This process of identifying visual text at a sentence level is what our work contributes to, which is an orthogonal first step to leverage the capabilities of text-to-image generation (and retrieval) models in a _targeted_ manner. Therefore, we disagree with the reviewer’s assessment that the work needs to demonstrate the increased quality of images generated by text-to-image models. Using qualitative analysis, we demonstrate that separating visual and non-visual text as the first step could avoid generating absurd images for non-visual text (Figures 4 and 6).
>
> **CLIP as a baseline**: The reviewer mentions their concern about using pre-trained CLIP as one of the baselines. Given the training objective, the score $1 - <I_{null}, T>$ demonstrates how dissimilar text $T$ is from the $NULL$ image, which is being used as a measure of its visualness. As such, there is no requirement for this score to correlate with $<I_{actual}, T>$, where $I_{actual}$ represents the true image corresponding to visual text $T$. This is what the reviewer expresses their concern about. In fact, given the problem formulation, $I_{actual}$ is not known during inference time. During inference, the task is to categorize $T$ into visual or non-visual using the similarity with a static $NULL$ image. Once the text is identified as visual, we show in Section 6 (under ‘Effect on text-to-image retrieval’) that our proposed fine-tuning preserves the retrieval capabilities of the CLIP model.
>
> We request the reviewer to reconsider their scores based on these clarifications.

---

### Meta-Review · Area_Chair_HLCp · 2023-09-12

**Recommendation:** 4

**Metareview:**

This paper looks at the task of determining if a give phrase or text evokes visual imagery or not. They motivate this by stating that understanding the visualness of text can be a precursor step to help generation models  and retrieval models (focus on just the visual text). They curate a dataset of 3620 sentences for this task, and fine tune large vision language models like CLIP to identify visual text.

The reviewers were able to understand the contributions of the paper and 2 of 4 reviewers found it exciting, and all reviewers thought at least the major claims in the paper supported sufficiently (soundness=3). There were some clarifications regarding better ways of quantifying the visualness, and also corrections in the baseline formulations that the reviewers suggested. The authors have responded to those, and indicated they will correct their work based on the feedback. Overall the paper will improve and benefit from the feedback suggested by the reviewers.

---

### Decision · Program_Chairs · 2023-10-07

**Decision:**

Accept-Main

**Comment:**

This paper looks at the task of determining if a give phrase or text evokes visual imagery or not. They motivate this by stating that understanding the visualness of text can be a precursor step to help generation models  and retrieval models (focus on just the visual text). They curate a dataset of 3620 sentences for this task, and fine tune large vision language models like CLIP to identify visual text.

The reviewers were able to understand the contributions of the paper and 2 of 4 reviewers found it exciting, and all reviewers thought at least the major claims in the paper supported sufficiently (soundness=3). There were some clarifications regarding better ways of quantifying the visualness, and also corrections in the baseline formulations that the reviewers suggested. The authors have responded to those, and indicated they will correct their work based on the feedback. Overall the paper will improve and benefit from the feedback suggested by the reviewers.